# Geometric Modeling for Control of Thermodynamic Systems

**DOI:** 10.3390/e25040577

**Published:** 2023-03-27

**Authors:** Arjan van der Schaft

**Affiliations:** Bernoulli Institute for Mathematics, Computer Science and Artificial Intelligence, Jan C. Willems Center for Systems and Control, University of Groningen, 9747 AG Groningen, The Netherlands; a.j.van.der.schaft@rug.nl

**Keywords:** macroscopic thermodynamics, dissipativity theory, Liouville geometry, homogeneous Hamiltonian dynamics, interconnection, control

## Abstract

This paper discusses the way that energy and entropy can be regarded as storage functions with respect to supply rates corresponding to the power and thermal ports of the thermodynamic system. Then, this research demonstrates how the factorization of the irreversible entropy production leads to quasi-Hamiltonian formulations, and how this can be used for stability analysis. The Liouville geometry approach to contact geometry is summarized, and how this leads to the definition of port-thermodynamic systems is discussed. This notion is utilized for control by interconnection of thermodynamic systems.

## 1. Introduction

Since the early 1970s [1] *contact geometry* has been recognized as underlying macroscopic thermodynamics, starting from Gibbs’ fundamental thermodynamic relation. This has spurred a series of papers on the geometry of thermodynamics; including [2,3,4,5,6,7,8,9,10,11,12,13,14,15,16,17,18,19,20,21,22,23,24,25,26,27]; see [28] for an introduction and survey. Nevertheless, this literature points to major differences with, for example, the geometric theory of classical mechanics (using symplectic geometry), and hints at aspects which have not yet been addressed. First, the thermodynamic phase space (which is formulated as a contact manifold) comprises the extensive *and* intensive variables, and thus, its dimension is more than twice the minimal number of variables to describe the thermodynamic system at any moment of time. Second, most of the theory is about *thermostatics*, and the proper geometric formulation of the dynamics is much less clear. Third, the contact geometric approach to thermodynamics is usually based on the *energy representation* of thermodynamic systems and its corresponding Gibbs one-form. On the other hand, there is also the *entropy formulation* which corresponds to another (although conformally equivalent) one-form. This led [29] to the use of homogeneous coordinates for the intensive variables, and thus to extend the thermodynamic phase space by one more degree of freedom. This was followed up in [25,26] by emphasizing the formulation of the thermodynamic phase space as the projection of the cotangent bundle over the space of extensive variables. Thus contact geometry is approached from the vantage point of the geometry of cotangent bundles with their Liouville one-form. Fourth, until now, not much work has been performed regarding the geometry of *irreversible thermodynamics*, based on the factorization of the irreversible entropy production. Fifth, how to use these geometric frameworks for the *control* of thermodynamic systems has not yet been addressed.

The present paper continues the investigation of all of these aspects. In Section 2, a systems and control perspective on macroscopic thermodynamics is emphasized by primarily regarding thermodynamic systems as systems interacting with their surroundings via heat, mechanical work, exchange of chemical species, etc. A classical example is, of course, the heat engine. A summary of how dissipativity theory provides a natural framework for interpreting and formulating the first and second laws of thermodynamics, Clausius’ inequality, and eventually entropy is provided. Indeed, energy and entropy reveal themselves to be the storage functions corresponding to two supply rates involving the thermal and mechanical ports of the thermodynamic system. Finally, this leads to Gibbs’ fundamental relation and to the definition of the thermodynamic phase space. Section 3 focuses on geometric descriptions of irreversible thermodynamic systems. The way that the classical factorization of the irreversible entropy production suggests quasi-Hamiltonian formulations (somewhat resembling GENERIC [30]) based on energy conservation and the increase of entropy of the autonomous part of the dynamics are discussed. This paper also indicates how such formulations may be used for stability analysis. Section 4 starts with the geometry of the thermodynamic phase space from the point of view of the Liouville geometry of the cotangent bundle over the space of extensive variables. Identifying the constitutive relations (‘thermostatics’) of the thermodynamic system as a Liouville submanifold, and the dynamics as homogeneous Hamiltonian dynamics lead to the definition of a port-thermodynamic system. Such systems interact with their environment via power ports and/or entropy flow ports. In Section 5, this is used for ‘control by interconnection’ of port-thermodynamic systems, where the dynamics of the system are sought to be controlled by interconnection with a suitable controller port-thermodynamic system. Finally, Section 6 contains conclusions and a discussion of venues for further research.

The sections are illustrated by three running examples: the gas-piston-damper system, chemical reaction networks, and the heat exchanger. Overall, the paper heavily builds upon previous papers [25,26,31,32,33,34], in which further details and background can be found.

## 2. The First and Second Law from the Point of View of Dissipativity Theory

The first law of thermodynamics expresses two fundamental properties: (1) the different types of interaction of a thermodynamic system with its surroundings (e.g., heat flow, mechanical work, flow of chemical species, etc.) all result in an exchange of a common quantity called *energy*, (2) there exists a function of the macroscopic thermodynamic variables that represents the energy *stored* in the system, and the increase of this function during any time interval is equal to the sum of the energies supplied to the system during this time interval by its surroundings (*conservation of energy*). Thus, energy manifests itself in different physical forms, which are *equivalent* and to a certain extent *exchangeable*. ‘To a certain extent’ because, as expressed by the second law of thermodynamics, there are *limitations* to the conversion of heat to other forms of energy.

The first law can be mathematically formulated through the use of *dissipativity theory* as formulated in [35]; see also [31,36,37]. Consider a simple thermodynamic system such as a gas, described by three variables: volume *V*, pressure *P*, and temperature *T*. Then, mechanical power (rate of mechanical work) provided by the surroundings to the thermodynamic system is given by $-Pu_{V}$, where $u_{V}:=\dot{V}$ is the rate of volume change. (In the physics convention for the pressure *P*, $Pu_{V}$ is the rate of mechanical work exerted by the system *on* the surroundings). The second type of interaction with the surroundings comes from heat delivered to the system (for instance, from a heat source). Let us denote, using *q*, the heat flow (heat per second) from the heat source into the system. Then the first law is expressed by the existence of a function E(x) of the thermodynamic state *x* (e.g., (V,P,T) satisfying the equation of state), expressing the energy of the system and satisfying, at all times, *t*
(1)$$\frac{d}{dt}E\left(x\left(t\right)\right)=q\left(t\right)-P\left(t\right)u_{V}\left(t\right)$$

That is, the increase of the total energy *E* of the thermodynamic system is equal to the incoming heat flow (through the thermal port) minus the mechanical work performed by the system on its surroundings (through the mechanical port). Equivalently, in the terminology of dissipativity theory, the first law amounts to the system being *cyclo-lossless* for the *supply rate* $q-Pu_{V}$, with *storage functionE*. This is directly extended to more involved thermodynamic systems. For example, suppose that apart from mechanical and thermal interaction with the surroundings, there is additional mass inflow of chemical species. Then, the supply rate $q-Pu_{V}$ is extended to $q-Pu_{V}+\sum_{k}\mu_{k}\nu_{k}$. Here, $\nu_{k}=\frac{dN_{k}}{dt}$, with $N_{k}$ the mole number of the *k*-th chemical species, and $\mu_{k}$ its chemical potential.

The first law emphasizes the role of thermodynamic systems as devices for *energy conversion*; energy from one physical domain is converted into energy in another domain. ‘Optimal’ conversion of heat into mechanical work, motivated by the design of steam engines in the beginning of the 19th century, was one of the starting points of thermodynamic theory. Electro-chemical devices such as batteries, and electro-mechanical systems including electrical motors and generators, are among the many other classical examples of energy-converting devices [38]. On the other hand, almost from the very start of thermodynamic theory, it was realized that there are intrinsic *limitations* to energy conversion. In particular, heat *cannot* just be converted into mechanical work. This is the origin of the second law of thermodynamics. The second law also admits a dissipativity interpretation; however, more involved than that of the first law. Let us start with the formulation of the second law, as given by Lord Kelvin (see [39]):


*A transformation of a thermodynamic system whose only final result is to transform into work heat extracted from a source which is at the same temperature throughout is impossible.*


Since the work done during a time interval $\left[t_{1},t_{2}\right]$ is equal to $\int_{t_{1}}^{t_{2}}-P\left(t\right)dV\left(t\right)=\int_{t_{1}}^{t_{2}}-P\left(t\right)u_{V}\left(t\right)dt$, Kelvin’s formulation immediately implies that for each *constant* temperature *T*, any thermodynamic system is *cyclo-passive* with respect to the supply rate $-Pu_{V}$. However, the second law is *stronger* than that. Namely, Kelvin’s formulation also forbids the conversion into work of heat from a source at constant temperature for all transformations in which the system interacts as well with a *second* heat source at *another* temperature, as long as the net heat taken from this second heat source is zero. As demonstrated by Carnot, the interaction with heat sources at *different* temperatures is crucial for the conversion of heat into mechanical energy. This led to the famous Carnot cycle which can be described as follows: consider a simple thermodynamic system, in particular, a fluid or gas in a confined space of a certain volume. Control of the system functions in two ways: (1) via *isothermal* transformations, where *heat* is supplied to, or taken from, the system at a constant temperature (classically described as the interconnection of the thermodynamic system with an infinite heat reservoir at the temperature of the isothermal process), (2) via *adiabatic* transformations, where the only interaction with the surroundings is via *work* supplied to, or taken from, the system (classically described by the movement of a piston that changes the volume of the system, with a pressure equal to the pressure of the gas). A *cycle* consists of two isothermal transformations and two adiabatic transformations: first, an isothermal transformation at temperature $T_{h}$ (‘hot’) takes the system from an initial state to another state, secondly, an adiabatic transformation lowers the temperature of the system to $T_{c}$ (‘cold’), thirdly, an isothermal transformation at temperature $T_{c}$ takes the system to a state from which, fourthly, an adiabatic transformation takes the system back to the original initial state; see Figure 1.

The cycle is called a *Carnot cycle* if it is *reversible*; i.e., can be traversed in the opposite direction as well.

**Remark 1.** 
*In the exposition of the Carnot cycle, often terminologies such as ‘infinitesimally slow’, ‘quasi-reversible’, ‘quasi-static’, etc., are used. This is largely with regard to the interaction of a system with its surroundings as being implemented by actual physical devices. For example, an isothermal transformation is viewed as the result of the ‘real’ physical action of a force exerted by a piston on the gas (implying that the pressure delivered by the piston could be different from the pressure of the gas). Furthermore, the system is considered to be in ‘real’ physical contact with a heat reservoir at a certain temperature (which could differ from that of the gas). In contrast in, e.g., electrical network theory and control theory the concept of an ‘ideal’ control action is employed, where, for instance, the pressure and the temperature are directly controlled.*


The heat delivered to the system during the first isothermal at temperature $T_{h}$ is denoted by $Q_{h}$, and during the second isothermal at $T_{c}$ by $Q_{c}$ (generally $Q_{c}$ is negative). Then, by the first law, since the final state is equal to the initial state, $Q_{h}+Q_{c}=W$, where W=∫PdV is the mechanical work that is done by the thermodynamic system on its surroundings.

By an intricate reasoning from [39], see also [31], Kelvin’s formulation of the second law yields for any cycle the fundamental inequality
(2)$$\frac{Q_{h}}{T_{h}}+\frac{Q_{c}}{T_{c}}\leq 0,$$
with *equality* in the case of a Carnot cycle. Furthermore, the reasoning can be extended to complex cycles, consisting of *n* isothermals at temperatures $T_{i}$ and absorbed heat quantities $Q_{i}$, i=1,2,⋯,n, interlaced by *n* adiabatics, leading to
(3)$$\sum_{i=1}^{n}\frac{Q_{i}}{T_{i}}\leq 0,$$
with equality in the case of reversibility. Finally, a slight extension (approximating continuous heat flow time-functions q(·) by step functions with step values $Q_{1},\cdots ,Q_{n}$) yields the celebrated *Clausius inequality*
(4)$$\oint \frac{q(t)}{T(t)}dt\leq 0$$
for all cyclic processes $\left(q(\cdot ),T(\cdot )\right)$ (where *q* is the heat flow into the thermodynamic system, and *T* is the temperature of the system), with equality
(5)$$\oint \frac{q(t)}{T(t)}dt=0$$
holding for all reversible cyclic processes (see [31] for details and refinements).

From the point of view of dissipativity theory [31,35] the Clausius inequality (Equation 4) is the same as *cyclo-dissipativity* of the thermodynamic system with respect to the *supply rate* $-\frac{q}{T}$. Thus, assuming reachability from and controllability from some ground state $x^{*}$ this means, see [40], that there exists a *storage function F* for the supply rate $-\frac{q}{T}$, that is $\frac{d}{dt}F\leq -\frac{q}{T}$. Hence S:=−F satisfies
(6)$$\frac{d}{dt}S\geq \frac{q}{T}$$

The function *S* was called ‘*entropy*’ by Clausius, from the Greek word τρoπη for ‘transformation’.

From the point of view of dissipativity theory, the storage function *F* need not be unique. In order to guarantee the uniqueness of *F* (modulo a constant), and therefore of the entropy *S*, we additionally assume [31,40] that, given some ground state, for every thermodynamic state there exists a reversible cyclic transformation through this state and the ground state satisfying
(7)$$\oint \frac{q(t)}{T(t)}=0$$

This uniqueness of *S* is, explicitly or implicitly, always assumed in expositions of macroscopic thermodynamics, and also in this paper.

**Remark 2.** 
*The dissipativity theory formulation of the second law already appears in [35], but under the additional assumption that F is nonnegative. In fact, in [35,37] there exists a nonnegative storage function for the supply rate $-\frac{q}{T}$ (and thus the system is dissipative instead of merely cyclo-dissipative) if and only if for all initial conditions x*

(8)
$$F_{a}\left(x\right)=\sup\int_{0}^{\tau }-\frac{q(t)}{T(t)}dt<\infty ,$$

*where the supremum is taken over all τ≥0 and all heat flow functions q(·) on the time interval [0,τ], and corresponding temperature profiles T(·) resulting from x(0)=x. Furthermore, if (Equation 8) holds, then $F_{a}\geq 0$ is minimal among all nonnegative storage functions. It follows that $S_{a}=-F_{a}\leq 0$ given by*

(9)
$$S_{a}\left(x\right)=\inf\int_{0}^{\tau }\frac{q(t)}{T(t)}dt>-\infty$$

*is maximal among all nonpositive storage functions. Since an arbitrary constant may be added to S while still satisfying (Equation 6), the assumption that S is nonpositive is equivalent to S being bounded from above. However, in many thermodynamic systems the entropy is not bounded from above. Thus thermodynamic systems are generally only cyclo-dissipative with respect to the supply rate $-\frac{q}{T}$, and not dissipative.*


### The Thermodynamic Phase Space and Gibbs’ Relation

The next step is now to *add* the energy and entropy as extra extensive variables to the description of the thermodynamic system. In order to illustrate this, consider a simple thermodynamic system, with extensive variable *V* (volume) and intensive variables P,T (pressure and temperature). The *equation of state* is an equation f(V,P,T)=0 for some scalar function *f*. (For example, for an ideal gas PV=RT with *R* the universal gas constant.) Any (V,P,T) satisfying f(V,P,T)=0 is called a *state* of the thermodynamic system. Hence, under regularity conditions the set of states of the thermodynamic system is a 2-dimensional *submanifoldM* of $R^{3}$. Then, consider the functions E:M→R (*energy*) and S:M→R (*entropy*) as obtained from dissipativity theory. Then, we may equally well represent the set of states $M\subset R^{3}$ by the 2-dimensional submanifold $L\subset R^{5}$ comprising the *total* set of extensive and intensive variables $R^{5}$:(10)L:={(E,S,V,T,P)∣f(V,P,T)=0,E=E(V,P,T),S=E(V,P,T)}

(With some abuse of notation, the extra *variables* E,S, are denoted by the same letters as used for the *functions* defined before.) The space $R^{5}$ of all extensive and intensive variables is called the *thermodynamic phase*.

Furthermore, by the first law $\frac{d}{dt}E=-P\frac{d}{dt}V+q$, while for any state there exists a path through this state and the ground state such that $\frac{d}{dt}E=-P\frac{d}{dt}V+T\frac{d}{dt}S$. Taken together, this implies that the *Gibbs one-form* on the thermodynamic phase space $R^{5}$ defined as
(11)dE−TdS+PdV,Gibbsone−form,
is *zero* restricted to *L*. This is called *Gibbs’ fundamental thermodynamic relation*. The thermodynamic phase space, together with the Gibbs one-form, defines a *contact manifold*. Furthermore, a submanifold of the thermodynamic phase space $R^{5}$ restricted to which the Gibbs one-form (Equation 11) is zero, and moreover has maximal dimension (in this case 2), is called a *Legendre submanifold*. Gibbs’ fundamental relation implies that any Legendre submanifold *L* is actually given as
(12)$$L:=\{\left(E,S,V,T,P\right)|E=E\left(S,V\right),T=\frac{\partial E}{\partial S}\left(S,V\right),-P=\frac{\partial E}{\partial V}\left(S,V\right)\}$$
for some energy functions E(S,V). Thus, *L* is completely described by expressing the energy *E* as a function E(S,V) of the other two extensive variables S,V, hence the name *energy representation*. Instead of relying on such an energy function (or its partial Legendre transforms), there is still an *alternative* way of describing *L*. This is to start, not with E(S,V), but instead with the expression of the *entropy* as a function S(E,V). For a simple thermodynamic system this leads to the *entropy representation* of the submanifold $L\subset R^{5}$ given as
(13)$$L:=\{\left(E,S,V,T,P\right)|S=S\left(E,V\right),\frac{1}{T}=\frac{\partial S}{\partial E}\left(E,V\right),\frac{P}{T}=\frac{\partial S}{\partial V}\left(E,V\right)\}$$

## 3. Irreversible Thermodynamics

Clausius interpreted the term $\frac{q}{T}$ in the inequality (Equation 6) as the part of the infinitesimal transformation $\frac{d}{dt}S$ that is *compensated* by the opposite rate of change $-\frac{q}{T}$ of the entropy of the *surroundings*; that is, of the reservoir supplying the heat to the thermodynamic system. The remaining part
(14)$$\sigma :=\frac{d}{dt}S-\frac{q}{T}\geq 0$$
was called the ‘uncompensated transformation’ by Clausius, and later the *irreversible entropy production* [38]. *Irreversible thermodynamics* is concerned with thermodynamics where σ is different from zero, implying an autonomous (independent from external heat flow) increase of the entropy *S*. Sometimes it is also referred to as *non-equilibrium thermodynamics*, because the entropy increase is resulting from (internal) non-equilibrium conditions.

The standard postulate of irreversible thermodynamics (see e.g., [38]) is that σ can be factorized as
(15)$$\sigma =\sum_{k=1}^{s}F_{k}J_{k}\geq 0,$$
where $F_{k}$ are called the *thermodynamic forces* and $J_{k}$ are the *thermodynamic flows* (or fluxes), in such a way that
(16)$$\sigma =0\;\mathrm{if}\;\mathrm{and}\;\mathrm{only}\;\mathrm{if}\;\;F_{k}=0,k=1,\cdots ,s$$

In *linear* irreversible thermodynamics [38] it is furthermore assumed that the vectors *F* and *J* with components $F_{k}$ and $J_{k},k=1,\cdots ,s,$ are related by a symmetric linear map
(17)$$J=LF,\;L=L^{⊤}$$

These are the celebrated *Onsager reciprocity relations* [38], corresponding to the symmetric factorization $\sigma =F^{⊤}LF$.

**Example** **1** (The heat exchanger)**.**
*The perhaps simplest example of irreversible dynamics and irreversible entropy production is offered by the heat exchanger. Consider two heat compartments, having temperatures*
$T_{h}$ and $T_{c}$
*(‘hot’ and ‘cold’), connected by a heat-conducting wall. In the absence of the conducting wall (and thus, without irreversible entropy production), these are two separate systems with entropies*
$S_{h}$
*and*
$S_{c}$*, each satisfying*
(18)$$\frac{d}{dt}S_{h}=\frac{q_{h}}{T_{h}},\;\frac{d}{dt}S_{c}=\frac{q_{c}}{T_{c}}$$
*Due to the conducting wall, there is a heat flow *q* from the hot to the cold compartment, which is given by Fourier’s law for heat conduction as*

$q=\lambda (T_{h}-T_{c})$

*for some positive constant*

λ

*. Furthermore, in view of the first law*

$q=-q_{h}=q_{c}$

*. Hence, the total entropy*

$S:=S_{h}+S_{c}$

*satisfies*

(19)
$$\frac{d}{dt}S=-\frac{q}{T_{h}}+\frac{q}{T_{c}}=\left(\frac{1}{T_{c}}-\frac{1}{T_{h}}\right)q$$


*This yields the following expression for the irreversible entropy production*

$\sigma =\frac{d}{dt}S$

*due to heat conduction (non-equilibrium conditions)*

(20)
$$\sigma =\left(\frac{1}{T_{c}}-\frac{1}{T_{h}}\right)\lambda \left(T_{h}-T_{c}\right)=\lambda \frac{{\left(T_{h}-T_{c}\right)}^{2}}{T_{h}T_{c}}\geq 0$$

*In this example the thermodynamic force is*$F=\frac{1}{T_{c}}-\frac{1}{T_{h}}$*, while the thermodynamic flow is*$J=q=\lambda (T_{h}-T_{c})$*. Indeed*σ=0*if, and only if,*F=0*. Despite its simplicity, this is an example of nonlinear irreversible thermodynamics, since the thermodynamic flow*$q=\lambda (T_{h}-T_{c})$*cannot be expressed as a linear function of the thermodynamic force*$F=\frac{1}{T_{c}}-\frac{1}{T_{h}}$.

**Example** **2** (The gas-piston-damper system)**.**
*Another simple example is the gas-piston-damper system. Consider a cylinder containing a gas whose volume can be controlled by a piston actuated by an external force u, and is subject to linear damping. The total energy E of the system can be expressed as a function of the other extensive variables as*
(21)$$E\left(S,V,\pi \right)=U\left(S,V\right)+\frac{\pi^{2}}{2m},$$
*with S representing entropy, V volume,*
π
*momentum of the piston with mass m, and*
U(S,V)
*representing the internal energy of the gas. Assuming that the heat as produced by the damping of the piston is fully absorbed by the gas in the cylinder, the dynamics are given as*
(22)$$\begin{array}{ccc} \dot{V} & = & A\frac{\pi }{m}\\ \dot{\pi } & = & -A\frac{\partial U}{\partial V}-dv+u\\ \dot{S} & = & \frac{dv^{2}}{T} \end{array}$$
*with*
$v=\frac{\pi }{m}$
*the velocity of the piston, A its area, d the damping constant,*
$T=\frac{\partial U}{\partial S}$
*the temperature, and u the external force on the piston. The thermodynamic force F is identified as*
$\frac{dv}{T}$
*and the thermodynamic flow as*
J=v*. Clearly Onsager relations*
J=LF
*are satisfied with*
$L=\frac{T}{d}$.

**Example** **3** (Chemical reaction network)**.**
*A third, more involved, example of irreversible thermodynamics are the dynamics of chemical reaction networks [34,41]. Consider an isolated (no incoming or outgoing chemical species, and no external heat flow) reaction network, with m chemical species and r reactions. Disregarding volume and pressure, consider the vector*
$x\in R^{m}$
*of concentrations of the chemical species. The dynamics take the form*
(23)$$\dot{x}=Nv\left(x\right),$$
*where*
$v\in R^{r}$
*is the vector of reaction fluxes. The*
m×r
*stoichiometric matrixN, which consists of positive and negative integer elements, captures the structural balance laws of the reactions. Chemical reaction network theory, as originating from [42,43,44], identifies the edges of the underlying directed graph with the r reactions, and the nodes with the ccomplexes of the chemical reactions, i.e., the different left- and right-hand sides of the reactions in the network. This means that the stoichiometric matrix N is factorized as*
N=ZB*, with B denoting the*
c×r
*incidence matrix of the graph of complexes, and Z denoting the*
m×c
*complex composition matrix (a matrix of nonnegative integers), whose*
ρ*-th column captures the expression of the*
ρ*-th complex in the m chemical species. It is shown in [45] that the dynamics*
$\dot{x}=Nv\left(x\right)$
*of a large class of chemical reaction networks (including detailed-balanced mass action kinetics networks) can be written into the compact form*
(24)$$\dot{x}=Nv\left(x\right)=-ZL\;\mathrm{Exp}\left(\frac{1}{RT}Z^{⊤}\mu \left(x\right)\right),$$
*where*
Exp
*is the vector exponential mapping*
$\;\mathrm{Exp}\left(z\right)={\left(\exp z_{1},\cdots ,\exp z_{c}\right)}^{⊤}$, *R is the gas constant, T is the temperature, and*
μ
*is the m-dimensional vector of chemical potentials of the chemical species (for which e.g., in the case of detailed-balanced mass action kinetics explicit expressions are available). Furthermore, the matrix*
$L:=BKB^{⊤}$
*in (Equation 24) defines a weighted Laplacian matrix for the graph of complexes, with the diagonal elements*
$\kappa_{1},\cdots ,\kappa_{r}$
*of the diagonal matrix*
K*, depending on the temperature T and the reference state. We have the following fundamental property [45]*
(25)$$\gamma^{⊤}L\;\mathrm{Exp}\;\gamma \geq 0\mathrm{for}\;\mathrm{all}\;\gamma \in R^{c},\;\gamma^{⊤}L\;\mathrm{Exp}\;\gamma =0\mathrm{iff}B^{⊤}\gamma =0$$*Expressing the entropy S as a function of x and the total energy E, Gibbs’ fundamental relation yields*$\frac{\partial S}{\partial x}\left(x,E\right)=-\frac{\mu }{T},\frac{\partial S}{\partial E}\left(x,E\right)=\frac{1}{T}$*. This implies*(26)$$\frac{d}{dt}S=\frac{1}{T}\mu^{⊤}ZL\;\mathrm{Exp}\left(\frac{Z^{⊤}\mu }{RT}\right)=:\sigma \geq 0,$$*with equality if, and only if,*$B^{⊤}Z^{⊤}\mu =N^{⊤}\mu =0$*, i.e., if and only if the chemical affinities*$N^{⊤}\mu$*of the reactions are all zero. Hence the equilibria of the system correspond to states of minimal (i.e., zero) entropy production*σ*, in accordance with the theory of irreversible thermodynamics [38]*.*The vectors of thermodynamic forces F and thermodynamic flows J are given as*(27)$$F=\frac{1}{T}N^{⊤}\mu ,\;J=KB^{⊤}\;\mathrm{Exp}\frac{Z^{⊤}\mu }{RT},$$*and indeed by (Equation 25)*σ=0*if and only if*F=0. *Note that Jcannot be expressed as a linear function of F and thus, in general, chemical reaction networks define nonlinear irreversible thermodynamics.*

### 3.1. Quasi-Hamiltonian Formulation of Irreversible Thermodynamic Systems

Conservative mechanical systems are well-known to admit a Hamiltonian formulation. The same holds for many other physical systems. The Hamiltonian formulation of the dynamics of thermodynamic systems is, however, much more elusive. This has already studied and elaborated upon in, e.g., [16,24,41,46]. The present formulation emphasizes the factorization (Equation 15) of the irreversible entropy production.

Consider an isolated thermodynamic system with entropy *S* and energy *E*. Collect all other extensive variables in a vector denoted by *z*. The energy *E* can be expressed as a function E=E(z,S) of *z* and *S*. Now consider the irreversible entropy production $\dot{S}=\sigma =J^{⊤}F$, with *J* the vector of thermodynamic flows and *F* the vector of thermodynamic forces. Often (as illustrated by the examples to be discussed), the thermodynamic force *F* can be expressed as $C^{⊤}\frac{\partial E}{\partial z}$ for some matrix *C*, whose elements are possibly depending on $\frac{\partial E}{\partial z},\frac{\partial E}{\partial S}$, as well as on z,S. Note that $\frac{\partial E}{\partial z}$ equals the vector of intensive variables associated with the extensive variables *z*, while the intensive variable $\frac{\partial E}{\partial S}$ equals the temperature *T*.

Energy conservation $\frac{d}{dt}E\left(z,S\right)=0$ together with $\frac{d}{dt}S\left(z,E\right)=J^{⊤}F$ suggests writing the dynamics of *z* and *S* into the form
(28)$$\left[\begin{array}{c} \dot{z}\\ \dot{S} \end{array}\right]=\underset{J_{e}}{\underset{︸}{\left[\begin{array}{cc} J & -CJ\\ J^{⊤}C^{⊤} & 0 \end{array}\right]}}\left[\begin{array}{c} \frac{\partial E}{\partial z}\\ \frac{\partial E}{\partial S} \end{array}\right],$$
for some skew-symmetric matrix J, possibly depending on $\frac{\partial E}{\partial z},\frac{\partial E}{\partial S}$ and z,S. This implies that the extended matrix $J_{e}$ is also skew-symmetric, and thus indeed $\frac{d}{dt}E\left(z,S\right)=0$. Note however, that since the matrix $J_{e}$ may depend on the *intensive* variables $\frac{\partial E}{\partial z},\frac{\partial E}{\partial S}$, it does *not* define a Poisson bracket on the state space with coordinates z,S. Therefore (Equation 28) will be called a *quasi*-Hamiltonian formulation.

This is illustrated by the previously discussed examples of the gas-piston-damper system, chemical reaction network, and heat exchanger as follows.

**Example** **4** (Gas-piston-damper system continued)**.**
*The dynamics of the gas-piston-damper system (Equation 22) can be written into the quasi-Hamiltonian form as (see also [41])*
(29)$$\left[\begin{array}{c} \dot{V}\\ \dot{\pi }\\ \dot{S} \end{array}\right]=\underset{J_{e}}{\underset{︸}{\left[\begin{array}{ccc} 0 & A & 0\\ -A & 0 & -d\frac{v}{T}\\ 0 & d\frac{v}{T} & 0 \end{array}\right]}}\left[\begin{array}{c} \frac{\partial E}{\partial V}\\ \frac{\partial E}{\partial \pi }\\ \frac{\partial E}{\partial S} \end{array}\right]+\left[\begin{array}{c} 0\\ 1\\ 0 \end{array}\right]u$$
*with*
$v=\frac{\partial E}{\partial \pi }=\frac{\pi }{m}$
*as the velocity of the piston and*
$T=\frac{\partial E}{\partial S}$
*the temperature. The thermodynamic flow and force are*
J=v and $F=\frac{d}{T}v$*, respectively. Hence,*
$J_{e}$
*is of the form as given in (Equation 28) with*
$C^{⊤}=\left[\begin{array}{cc} 0 & \frac{d}{T} \end{array}\right]$. $J_{e}$
*depends on the intensive variables T and v, therefore, it does not define a Poisson bracket*.

**Example** **5** (Chemical reaction network continued)**.**
*In the case of chemical reaction networks, the vector of thermodynamic forces is given as*
$F=\frac{1}{T}N^{⊤}\mu =C^{⊤}\frac{\partial E}{\partial x}$
*with*
$C=\frac{1}{T}N$*, and*
$\frac{\partial E}{\partial x}=\mu$
*the vector of chemical potentials. Furthermore, according to (Equation 27) the vector of thermodynamic flows is given as*
$J=KB^{⊤}\;\mathrm{Exp}\frac{Z^{⊤}\mu }{RT}$*. This leads to the quasi-Hamiltonian representation*
(30)$$\left[\begin{array}{c} \dot{x}\\ \dot{S} \end{array}\right]=\left[\begin{array}{cc} 0 & -\frac{1}{T}NJ\\ \frac{1}{T}J^{⊤}N^{⊤} & 0 \end{array}\right]\left[\begin{array}{c} \frac{\partial E}{\partial x}\\ \frac{\partial E}{\partial S} \end{array}\right]$$

**Example** **6** (Heat exchanger continued)**.**
*The quasi-Hamiltonian formulation of the heat exchanger is slightly different. This caused by the fact that, in this example, we have two entropies,*
$S_{1}$
*and*
$S_{2}$*, corresponding to the two compartments (and not a total entropy as in the previous two examples). In fact, the quasi-Hamiltonian formulation of the heat exchanger is given as (see [41])*
(31)$$\left[\begin{array}{c} {\dot{S}}_{1}\\ {\dot{S}}_{2} \end{array}\right]=\left[\begin{array}{c} -\frac{q}{T_{1}}\\ \frac{q}{T_{2}} \end{array}\right]=\left[\begin{array}{cc} 0 & \lambda \left(\frac{1}{T_{1}}-\frac{1}{T_{2}}\right)\\ -\lambda \left(\frac{1}{T_{1}}-\frac{1}{T_{2}}\right) & 0 \end{array}\right]\left[\begin{array}{c} T_{1}\\ T_{2} \end{array}\right]$$
*since*
$\frac{\partial E}{\partial S_{i}}=T_{i},i=1,2$*, and the heat flow from compartment 1 to 2 is given by*
$q=\lambda (T_{1}-T_{2})$*. Here, we recognize*
$\frac{1}{T_{1}}-\frac{1}{T_{2}}$
*as the thermodynamic force*.

A further structured form of quasi-Hamiltonian modeling of irreversible thermodynamic systems, called *irreversible port-Hamiltonian systems*, was introduced in [24]; see [41,46,47] for more developments and references.

A special case occurs if the total energy E(z,S) splits as
(32)E(z,S)=H(z)+U(S),
for some thermal energy U(S) and remaining energy H(z). In this case, one obtains the equations
(33)$$\dot{z}=J\frac{\partial H}{\partial z}\left(z\right)-TCJ,\;\dot{S}=J^{⊤}C^{⊤}\frac{\partial H}{\partial z}\left(z\right),\;F=C^{⊤}\frac{\partial H}{\partial z}\left(z\right)$$

If, furthermore, J=LF with $L=L^{⊤}$ (Onsager’s reciprocity relations) then the dynamical equations for the extensive variables *z* can be combined into
(34)$$\dot{z}=J\frac{\partial H}{\partial z}\left(z\right)-TCLC^{⊤}\frac{\partial H}{\partial z}\left(z\right)=\left(J-TCLC^{⊤}\right)\frac{\partial H}{\partial z}\left(z\right)$$

This is the standard internal dynamics of a *port-Hamiltonian system* with state vector *z*; see e.g., [37,48,49]. In this case, irreversibility means that, even though the total energy E(z,S)=H(z)+U(S) is preserved, the part of the energy given by H(z) is continuously transformed (by the resistive power flow $T\dot{S}$) into the thermal energy U(S). Conversely, one can show [31] that any port-Hamiltonian system can be embedded into an energy-conserving thermodynamic system.

**Example 7** (Mass-spring-damper system)**.**
*A simple example is the ubiquitous mass-spring-damper system. Its dynamics are very similar to that of the gas-piston-damper system, the difference being that the internal energy*
U(V,S)
*of the gas is replaced by the sum*
$\frac{1}{2}kx^{2}+U\left(S\right)$, *where*
$\frac{1}{2}kx^{2}$
*is the potential energy of the spring (with x denoting the elongation of the spring), and*
U(S)
*is the thermal energy of the system. This leads to the dynamics (compare with (Equation 29))*
(35)$$\left[\begin{array}{c} \dot{x}\\ \dot{\pi }\\ \dot{S} \end{array}\right]=\left[\begin{array}{ccc} 0 & 1 & 0\\ -1 & 0 & -d\frac{\pi }{mT}\\ 0 & d\frac{\pi }{mT} & 0 \end{array}\right]\left[\begin{array}{c} kx\\ \frac{\pi }{m}\\ T \end{array}\right]+\left[\begin{array}{c} 0\\ 1\\ 0 \end{array}\right]u,$$
*as well as the following port-Hamiltonian formulation of the mass-spring-damper system*
(36)$$\left[\begin{array}{c} \dot{x}\\ \dot{\pi } \end{array}\right]=\left[\begin{array}{cc} 0 & 1\\ -1 & -d \end{array}\right]\left[\begin{array}{c} kx\\ \frac{\pi }{m} \end{array}\right]+\left[\begin{array}{c} 0\\ 1 \end{array}\right]u$$

### 3.2. Stability Analysis

The quasi-Hamiltonian formulation can be readily used for *stability analysis*. Note, however, that the conditions $\frac{\partial E}{\partial z}=0$, $\frac{\partial E}{\partial S}=0$ for *E* having a minimum often do not correspond to equilibria of interest. This is illustrated by the gas-piston-damper system, where these conditions correspond to pressure, velocity, and temperature all being equal to zero. Instead, in such cases it is of much more interest to consider the stability of *steady states*$\left(\bar{V},\bar{\pi }=0,\bar{S}\right)$ corresponding to a non-zero force $\bar{u}$ delivered by the piston. In view of (Equation 29) and the energy expression $E\left(V,\pi ,S\right)=\frac{\pi^{2}}{2m}+U\left(V,S\right)$, this corresponds to the steady state condition
(37)$$A\frac{\partial U}{\partial V}\left(\bar{V},\bar{S}\right)=\bar{u}$$
(Note that $\dot{S}=0$ is ensured by $\bar{\pi }=0$ implying that $\bar{v}=\frac{\bar{\pi }}{m}=0$, and thus corresponds to a singularity in the skew-symmetric matrix $J_{e}$, instead of a vanishing of all the partial derivatives of *E*. In particular, the temperature $T=\frac{\partial E}{\partial S}$ at steady state will not be zero.) Instead of using E(z,S) as a candidate Lyapunov function which leads to the consideration of the *availability function* [50] (also called Bregman divergence or shifted Hamiltonian [37])
(38)$$\hat{E}\left(V,\pi ,S\right):=\frac{\pi^{2}}{2m}+U\left(V,S\right)+\bar{P}\left(V-\bar{V}\right)-\bar{T}\left(S-\bar{S}\right)-E\left(\bar{V},\bar{\pi },\bar{S}\right),$$
where $P=-\frac{\partial U}{\partial V}\left(\bar{S},\bar{V}\right)$ and $\bar{T}=\frac{\partial U}{\partial S}\left(\bar{S},\bar{V}\right)$ are the pressure and temperature at steady state, for some arbitrary value $\bar{S}$. Indeed, using the steady state condition (Equation 37), a direct computation yields
(39)$$\frac{d}{dt}\hat{E}=-\frac{\bar{T}}{T}dv^{2}\leq 0$$
for all values of the temperature T>0 and the steady state temperature $\bar{T}>0$. Furthermore, given that for thermodynamic systems the internal energy U(V,S) (and therefore E(V,π,S)) is a *convex* function, $\hat{E}\left(V,\pi ,S\right)$ is also convex with minimum at $\bar{V},\bar{\pi }=0,\bar{S}$. Hence, if $\hat{E}\left(V,\pi ,S\right)$ is strictly convex (which is often the case), then this proves the asymptotic stability of the steady state. The use of the availability function for stability analysis and stabilization was already advocated for in [51]; see also, e.g., [47,52] for related work using the availability function in the context of passivity-based control of irreversible port-Hamiltonian systems.

This is extended to *general* quasi-Hamiltonian systems
(40)$$\left[\begin{array}{c} \dot{z}\\ \dot{S} \end{array}\right]=J_{e}\left[\begin{array}{c} \frac{\partial E}{\partial z}\left(z,S\right)\\ \frac{\partial E}{\partial S}\left(z,S\right) \end{array}\right]+Gu,$$
where the skew–symmetric matrix $J_{e}$ and the input matrix *G* may both depend on the extensive variables z,S and the intensive variables $\frac{\partial E}{\partial z}\left(z,S\right),\frac{\partial E}{\partial S}\left(z,S\right)=T$. The steady state condition for $u=\bar{u}$ is given as
(41)$${\bar{J}}_{e}\left[\begin{array}{c} \frac{\partial E}{\partial z}\left(\bar{z},\bar{S}\right)\\ \frac{\partial E}{\partial S}\left(\bar{z},\bar{S}\right) \end{array}\right]+\bar{G}\bar{u}=0,$$
where ${\bar{J}}_{e}$ and $\bar{G}$ denote the values of $J_{e}$ and *G* at steady state, i.e.,
(42)$${\bar{J}}_{e}=J_{e}\left(\bar{z},\bar{S},\frac{\partial E}{\partial z}\left(\bar{z},\bar{S}\right),\frac{\partial E}{\partial S}\left(\bar{z},\bar{S}\right)\right),\;\bar{G}=G\left(\bar{z},\bar{S},\frac{\partial E}{\partial z}(\bar{z},\bar{S},\frac{\partial E}{\partial S}\left(\bar{z},\bar{S}\right)\right)$$

Assuming the energy function E(z,S) to be *convex* (which is normally the case in thermodynamic systems), then the availability function is given as the convex function
(43)$$\hat{E}\left(z,S\right):=E\left(z,S\right)-\frac{\partial E}{\partial z^{⊤}}\left(\bar{z},\bar{S}\right)\left(z-\bar{z}\right)-\frac{\partial E}{\partial S}\left(\bar{z},\bar{S}\right)\left(S-\bar{S}\right)-E\left(\bar{z},\bar{S}\right),$$
having a minimum at $\left(\bar{z},\bar{S}\right)$. A key property of the availability function $\hat{E}$ is that
(44)$$\nabla \hat{E}\left(z,S\right)=\nabla E\left(z,S\right)-\nabla E\left(\bar{z},\bar{S}\right)$$
where $\nabla \hat{E}\left(z,S\right)$ denotes the gradient vector of $\hat{E}$ (written as a column vector). The computation of $\frac{d}{dt}\hat{E}\left(z,S\right)$ yields, exploiting the steady state condition (Equation 41),
(45)$$\frac{d}{dt}\hat{E}\left(z,S\right)={\left(\nabla \hat{E}\left(z,S\right)\right)}^{⊤}\left[J_{e}\nabla E\left(z,S\right)-{\bar{J}}_{e}\nabla E\left(\bar{z},\bar{S}\right)\right]$$

It follows that $\frac{d}{dt}\hat{E}\left(z,S\right)\leq 0$ if, and only if
(46)$${\left(\nabla E\left(z,S\right)-\nabla E\left(\bar{z},\bar{S}\right)\right)}^{⊤}\left[J_{e}\nabla E\left(z,S\right)-{\bar{J}}_{e}\nabla E\left(\bar{z},\bar{S}\right)\right]\leq 0$$
(This condition is similar to the condition for asymptotic stability of steady states of port-Hamiltonian systems as derived in [53]; see also [37].) Hence if (Equation 46) is satisfied and $\hat{E}\left(z,S\right)$ is not only convex but even strictly convex, then $\hat{E}\left(z,S\right)$ serves as a Lyapunov function for assessing the (asymptotic) stability of the steady state $\left(\bar{z},\bar{S}\right)$.

Instead of expressing the energy *E* as a function E(z,S) of the remaining extensive variables z,S and writing the dynamics as a quasi-Hamiltonian system with Hamiltonian given by *E*, one may also write the entropy *S* as a function S(z,E) and try to express the dynamics as being generated by the gradient of this entropy function. However, since (in the isolated case) $\frac{d}{dt}S\geq 0$, this constitutes quite a different scenario. An example where it *is* possible is a chemical reaction network as mentioned before. Instead of the quasi-Hamiltonian formulation (Equation 30), one rewrites the dynamics as (with *z* replaced by the vector of concentrations *x*)
(47)$$\left[\begin{array}{c} \dot{x}\\ \dot{E} \end{array}\right]=\underset{F_{e}}{\underset{︸}{T\left[\begin{array}{cc} 0 & -NJ\\ J^{⊤}N^{⊤} & TJ^{⊤}F \end{array}\right]}}\left[\begin{array}{c} \frac{\partial S}{\partial x}\left(x,E\right)\\ \frac{\partial S}{\partial E}\left(x,E\right) \end{array}\right],\;$$
where $\frac{\partial S}{\partial x}\left(x,E\right)=-\frac{\mu }{T}$ and $\frac{\partial S}{\partial E}\left(x,E\right)=\frac{1}{T}$. Consequently
(48)$$\dot{E}=-J^{⊤}N^{⊤}\mu +TJ^{⊤}F=0,$$
because of $F=\frac{1}{T}N^{⊤}\mu$. Since $\frac{d}{dt}S\geq 0$, the availability function corresponding to $V\left(z\right):=-S\left(x,\bar{E}\right)$, with $\bar{E}$ the constant total energy of the system, can be used as a Lyapunov function for stability analysis; cf. [34] for details. Note that the matrix $F_{e}$, in the right-hand of (Equation 47), is *not* a skew-symmetric matrix anymore. In fact, the formulation (Equation 47) resembles the formulation of thermodynamic systems as used in the GENERIC formalism; see, e.g., [30].

Another interesting case are *isothermal* chemical reaction networks. In this case [45] one considers the Gibbs free energy (Legendre transform of E(z,S) with respect to *S*)
(49)$$G\left(z,T\right)=E\left(z,S\right)-TS,\;T=\frac{\partial E}{\partial S}$$
for constant *T*. By the properties of the Legendre transform $\frac{\partial G}{\partial z}=\frac{\partial E}{\partial z}=\mu$ (the vector of chemical potentials). Hence, in view of (Equation 47) one obtains for constant *T*
(50)$$\frac{d}{dt}G=\mu^{⊤}\dot{z}=-\mu^{⊤}NJ=TF^{⊤}J=-T\sigma$$
with σ the irreversible entropy production. Alternatively expressed, whenever the temperature *T* is constant, $\frac{d}{dt}G=\frac{d}{dt}E-T\frac{d}{dt}S$, while $\frac{d}{dt}E=q$ (with *q* the heat flow needed to keep the temperature constant) and $\frac{d}{dt}S=q-\sigma$. Taken together this indeed yields $\frac{d}{dt}G=-T\sigma$.

## 4. Thermodynamic Phase Space and Liouville Geometry

A typical feature of thermodynamic systems modeling is the use of many more variables than the minimal number of variables to describe the ‘state’ of the system. For example, a simple thermodynamic system is described by a 2-dimensional submanifold *L* of the 3-dimensional space of macroscopic quantities V,P,T; one extensive, and two intensive. Then, based on the first and second laws of thermodynamics, two extra extensive variables E,S are introduced. As a result, the system is described as a 2-dimensional submanifold *L* of $R^{5}$; the thermodynamic phase space is generated by the three extensive variables E,S,V, and the two intensive variables T,P. This is immediately extended to the general thermodynamic case, where *L* is an *n*-dimensional submanifold of the (2n+1)-dimensional thermodynamic phase space (comprising n+1 extensive variables and *n* intensive variables).

### 4.1. Constitutive Relations and Liouville Submanifolds

The Legendre submanifold *L* only defines the *constitutive relations* of the thermodynamic system, i.e., its *thermostatics*. The first and second laws impose *constraints* on any possible dynamics of the thermodynamic system, but do *not* define it. On the other hand, two requirements for any dynamics on the full thermodynamic phase space are natural: (1) the dynamics should respect the ‘structure’ of the thermodynamic phase space, (2) it should respect the constitutive relations; i.e., should leave the submanifold *L invariant*. The proper geometry to address this is *contact geometry*. However, in order to unify the energy and entropy representation we will take one more abstraction step; from contact geometry to *Liouville geometry*. This will have the additional benefit of making a clear separation between extensive and intensive variables, and of being computationally more easy; see [25,26,54,55] for further details and ramifications.

For a simple thermodynamic system with extensive variables E,S,V and intensive variables T,−P, the step from contact to Liouville geometry amounts to replacing the intensive variables T,−P (in the energy representation) with their *homogeneous coordinates*$p_{E},p_{S},p_{V}$ with $p_{E}\neq 0$, i.e.,
(51)$$T=\frac{p_{S}}{-p_{E}},\;-P=\frac{p_{V}}{-p_{E}},$$
and thereby to express the intensive variables $\frac{1}{T},\frac{P}{T}$ in the entropy representation as
(52)$$\frac{1}{T}=\frac{p_{E}}{-p_{S}},\;\frac{P}{T}=\frac{p_{V}}{-p_{S}}$$

In this way, the *two* Gibbs one-forms dE−TdS+PdV and $dS-\frac{1}{T}dE-\frac{P}{T}dV$ are replaced by a *single* symmetric expression, namely by the *Liouville one-form*
(53)$$p_{E}dE+p_{S}dS+p_{V}dV,$$
on the cotangent bundle $T^{*}R^{3}$, with $R^{3}$ the space of extensive variables E,S,V. By definition of homogeneous coordinates, the vector $\left(p_{E},p_{S},p_{V}\right)$ is always different from the 0-vector. Hence, the space $\left\{\left(E,S,V,p_{E},p_{S},p_{V}\right)\right\}$ is actually the cotangent bundle $T^{*}R^{3}$ *minus* its zero section. Using homogeneous coordinates, the 2-dimensional Legendre submanifold *L* is now replaced by the 3-dimensional submanifold $L\subset T^{*}R^{3}$, given as
(54)$$L=\{\left(E,S,V,p_{E},p_{S},p_{V}\right)|\left(E,S,V,\frac{p_{S}}{-p_{E}},\frac{p_{V}}{-p_{E}}\right)\in L,\;\left(p_{E},p_{S},p_{V}\right)\neq 0\}$$

It turns out that L is a *Lagrangian submanifold* [56,57,58], which is moreover *homogeneous*, in the sense that whenever $(E,S,V,p_{E},p_{S},p_{V})\in L$ then also $(E,S,V,\lambda p_{E},\lambda p_{S},\lambda p_{V})\in L$, for any non-zero λ∈R. Such submanifolds are fully characterized as maximal manifolds restricted to which the Liouville form $p_{E}dE+p_{S}dS+p_{V}dV$ is zero, and are therefore called homogeneous Lagrangian submanifolds or *Liouville submanifolds* [25].

In general, one considers the (n+1)-dimensional manifold Q of all the extensive variables (including *E* and *S*), and its (2n+2)-dimensional cotangent bundle without zero section, denoted by $T^{*}Q$. The constitutive properties of the thermodynamic system are defined by a (n+1)-dimensional Liouville submanifold L. Conversely, starting from $T^{*}Q$ we may define a *contact manifold* in the following way [57]. For each q∈Q and cotangent space $T_{q}^{*}Q$ consider the *projective space*$P(T_{q}^{*}Q)$, given as the set of rays in $T_{q}^{*}Q$, that is, all the non-zero multiples of a non-zero cotangent vector. Thus, the projective space $P(T_{q}^{*}Q)$ has dimension *n*, and there is the canonical projection $\pi_{q}:T_{q}^{*}Q\to P\left(T_{q}^{*}Q\right)$, where $T_{q}^{*}Q$ denotes the cotangent space without its zero vector. The fiber bundle of the projective spaces $P(T_{q}^{*}Q)$, q∈Q, over the base manifold Q will be denoted by $P(T^{*}Q)$. Furthermore, the bundle projection obtained by considering $\pi_{q}:T_{q}^{*}Q\to P\left(T_{q}^{*}Q\right)$ for every q∈Q is denoted by $\pi :T^{*}Q\to P\left(T^{*}Q\right)$. As detailed in [26,57,58], $P(T^{*}Q)$ defines a contact manifold of dimension 2n+1, and will serve as the canonical *thermodynamic phase space* for the thermodynamic system with space of external variables Q. In the case of a simple thermodynamic system the bundle projection π is given in coordinates as
(55)$$\left(p_{E},p_{S},p_{V}\right)\mapsto \left(\frac{p_{S}}{-p_{E}}=T,\frac{p_{V}}{-p_{E}}=-P\right)$$
whenever $p_{E}\neq 0$ (energy representation), or as
(56)$$\left(p_{E},p_{S},p_{V}\right)\mapsto \left(\frac{p_{E}}{-p_{S}}=-\frac{1}{T},\frac{p_{V}}{-p_{S}}=\frac{P}{T}\right)$$
whenever $p_{S}\neq 0$ (entropy representation). The cotangent bundle $T^{*}Q$, and therefore also $T^{*}Q$, are endowed with the natural one-form α, called the *Liouville form*, which in natural cotangent bundle coordinates $\left(q,p\right)=\left(q_{1},\cdots ,q_{n+1},p_{1},\cdots ,p_{n+1}\right)$ is given as $\alpha =\sum_{i=1}^{n+1}p_{i}dq_{i}$. A submanifold $L\subset T^{*}Q$ is a Liouville submanifold if α restricted to L is zero, and furthermore L is maximal with respect to this property. It turns out that maximality of L is equivalent to dimL=n+1.

For any point in a Liouville submanifold L there exists a neighborhood of this point, a splitting of the index set {1,⋯,n+1}=I∪J and a function $F(q_{I},p_{J})$ that is homogeneous of degree 1 in $p_{J}$ (in particular J≠∅), where $q_{I}$ denotes the vector of coordinates $q_{i}$ with i∈I and $p_{J}$ denotes the vector of coordinates $p_{j}$ with j∈J, such that on this neighborhood
(57)$$L=\{\left(q,p\right)\in T^{*}Q|p_{i}=\frac{\partial F}{\partial q_{i}},i\in I,\;q_{j}=-\frac{\partial F}{\partial p_{j}},j\in J\}$$

By homogeneity of *F* in $p_{J}$ it follows that for any j∈J we can write, whenever $p_{j}\neq 0$,
(58)$$F\left(q_{I},p_{J}\right)=-p_{j}\hat{F}\left(q_{I},\frac{p_{\ell }}{-p_{j}}\right),\ell \in J,\ell \neq j)$$
for some function $\hat{F}$. The choice of j∈J corresponds to a choice of the coordinates for the contact manifold $P(T^{*}Q)$ (for example, corresponding to the energy or the entropy representation).

The Liouville submanifold L projects under π to a Legendre submanifold $L\subset P(T^{*}Q)$, and conversely any Legendre submanifold $L\subset P(T^{*}Q)$ is the projection of a Liouville submanifold $L\subset T^{*}Q$. Furthermore, the function $\hat{F}$ serves as a generating function for the Legendre submanifold *L*. Although the close relation of contact geometry with the Liouville geometry of cotangent bundles is well-known in differential geometry [57,58], the use of homogeneous coordinates for thermodynamics was first advocated for in [29].

### 4.2. Homogeneous Hamiltonian Dynamics and Port-Thermodynamic Systems

The dynamics of the thermodynamic system should now satisfy the following two basic requirements. First, it should respect the structure of $T^{*}Q$, and therefore of the contact manifold $P(T^{*}Q)$. Second, it should leave invariant the Liouville submanifold L specifying the constitutive relations. The first requirement amounts to a requirement that the dynamics are Hamiltonian on $T^{*}Q$, with the additional property that the Liouville forms on $T^{*}Q$ is preserved. This can be seen to correspond to Hamiltonian dynamics with a Hamiltonian *K* that is *homogeneous of degree 1* in the *p*-variables. Thus, if *q* are coordinates for *Q* and (q,p) are corresponding cotangent bundle coordinates for $T^{*}Q$ (such that the Liouville form is $\alpha =\sum_{i=1}^{n+1}p_{i}dq_{i}$), then we consider Hamiltonians K(q,p) satisfying K(q,λp)=λK(q,p) for all λ∈R different from zero. Equivalently, by Euler’s theorem, the Hamiltonian K(q,p) should satisfy
(59)$$K\left(q,p\right)=\sum_{i=1}^{n+1}p_{i}\frac{\partial K}{\partial p_{i}}\left(q,p\right),$$
with the functions $\frac{\partial K}{\partial p_{i}}\left(q,p\right)$ homogeneous of degree 0 in *p*, i=1,⋯,n+1. The second requirement is equivalent to *K* being such that *K* restricted to L is zero. Finally, we will split *K* into two parts, i.e.,
(60)$$K^{a}+K^{c}u,\;u\in R^{m}$$

Here $K^{a}:T^{*}Z\to R$ is the Hamiltonian corresponding to the autonomous dynamics due to internal non-equilibrium conditions, while $K^{c}=\left(K_{1}^{c},\cdots ,K_{m}^{c}\right)$ is a row vector of Hamiltonians corresponding to dynamics arising from interaction with the surroundings of the system, parameterized by a vector *u* of *control* or *input* variables. (However, as we will notice in the context of the *damper system* (Equation 81), there are cases where the dependence on *u* is not affine.) Thus all these Hamiltonians $K^{a},K_{1}^{c},\cdots ,K_{m}^{c}$ are zero on L. This implies that the dynamics of the extensive variables are given as
(61)$${\dot{q}}_{i}=\frac{\partial K^{a}}{\partial p_{i}}\left(q,p\right)+\sum_{j=1}^{m}\frac{\partial K^{c_{j}}}{\partial p_{i}}\left(q,p\right),\;i=1,\cdots ,n+1,$$
where the right-hand side is homogeneous of degree 0 in *p*. (Note that this does *not* mean that the right-hand side is necessarily independent of *p*; it may depend on degree 0 variables $\frac{p_{i}}{-p_{E}}$; i.e., on the intensive variables!)

Finally, there are two important *extra* constraints on $K^{a}$ (the Hamiltonian governing the autonomous dynamics corresponding to u=0) which are directly imposed by the first and second laws. By the first law $0=\dot{E}=\frac{\partial K^{a}}{\partial p_{E}}$ on L. Furthermore, by the second law necessarily $\frac{\partial K^{a}}{\partial p_{S}}{|}_{L}\geq 0$. In fact, using the postulate of factorization of the irreversible entropy production as discussed in Section 3 one has
(62)$$\frac{\partial K^{a}}{\partial p_{S}}{|}_{L}=\sigma =\sum_{k=1}^{s}F_{k}J_{k}\geq 0$$
where σ=0 if and only if $F_{k}=0,k=1,\cdots ,s$.

Such constraints do not hold for the control (interaction) Hamiltonians $K^{c}$. In fact, the corresponding terms of the control Hamiltonians define natural *outputs* conjugated to the inputs *u*. The first option is to define the *m*-dimensional row vector
(63)$$y_{p}=\frac{\partial K^{c}}{\partial p_{E}},$$
with the subscript *p* in $y_{p}$ standing for *power*. Then, it follows that $\frac{d}{dt}E=y_{p}u$, and thus, $y_{p}$ is the vector of *power-conjugate* (passive) outputs corresponding to the input vector *u*. Similarly, by defining the *m*-dimensional row vector
(64)$$y_{e}=\frac{\partial K^{c}}{\partial p_{S}}$$
it follows that $\frac{d}{dt}S\geq y_{e}u$. Hence $y_{e}$ is the output vector which is conjugate to *u* in terms of *entropy flow*. This is summarized in the following definition of a *port-thermodynamic system* as given in [25,26].

**Definition 1.** 
*Consider a manifold Q of extensive variables. A port-thermodynamic system on Q is defined by a Liouville submanifold $L\subset T^{*}Q$ specifying the constitutive relations of the system, together with a Hamiltonian $K^{a}+K^{c}u,u\in R^{m}$, homogeneous of degree 1 in p, which is zero on L for every u and satisfying $\frac{\partial K^{a}}{\partial p_{E}}=0$ on L, and $\frac{\partial K^{a}}{\partial p_{S}}\geq 0$ on L. Its power port is defined by u together with the output $y_{p}=\frac{\partial K^{c}}{\partial p_{E}}$, and its entropy flow port by u and $y_{e}=\frac{\partial K^{c}}{\partial p_{S}}$.*


**Remark 3.** 
*The Hamiltonian K generates the dynamics, but does not have an interpretation in terms of energy. In fact, K is dimensionless; see [25,26] for further information.*


**Remark 4.** 
*Through the use of entropy flow ports, one could express the irreversible entropy production $\sigma =\sum_{k=1}^{s}F_{k}J_{k}\geq 0$ as being the result of the interconnection of the system with a pure entropy producing element. This is especially clear if the vector of thermodynamic flows J can be expressed as a function of the vector of thermodynamic forces F, like in Onsager’s relations $J=LF,L=L^{⊤}$. Namely, in this case one may consider u=J and entropy conjugate outputs $y_{e}=F$, and then ‘close’ the loop by setting $u=Ly_{e}$.*


Because of the homogeneity of the Liouville submanifold L and of the Hamiltonian *K*, the port-thermodynamic system defined on $T^{*}Q$, including its power and entropy flow ports, *projects* to a system on the thermodynamic phase space $P(T^{*}Q)$ with Legendre submanifold *L*; see [25] for details. In fact, the resulting class of systems on $P(T^{*}Q)$ is very close to the classes of input–output contact systems and conservative control contact systems on contact manifolds as introduced and studied in [7,8,9,59,60].

The definition of port-thermodynamic systems (Definition 1) is illustrated by the examples of the gas-piston-damper system, chemical reaction network, and heat exchanger as follows.

**Example 8** (Gas-piston-damper system continued [26])**.**
*As discussed before, the extensive variables are*
*E*
*(energy)*, *S*
*(entropy)*, *V*
*(volume), and*
π
*(momentum of the piston). For simplicity we will take*
A=1. *The constitutive properties are given by the Liouville submanifold*
(65)$$\begin{array}{ccc} L & = & \{\left(E,S,V,\pi ,p_{E},p_{S},p_{V},p_{\pi }\right)|E\left(S,V,\pi \right)=U\left(S,V\right)+\frac{\pi^{2}}{2m},\\ & & p_{S}=-p_{E}\frac{\partial U}{\partial S},p_{V}=-p_{E}\frac{\partial U}{\partial V},p_{\pi }=-p_{E}\frac{\pi }{m}\} \end{array}$$
* with generating function*
$-p_{E}\left(U\left(S,V\right)+\frac{\pi^{2}}{2m}\right)$. *The dynamics are given by the Hamiltonian (homogeneous of degree 1 in*
*p*)
(66)$$K=p_{V}\frac{\pi }{m}+p_{\pi }\left(-\frac{\partial U}{\partial V}-d\frac{\pi }{m}\right)+p_{S}\frac{d{\left(\frac{\pi }{m}\right)}^{2}}{\frac{\partial U}{\partial S}}+\left(p_{\pi }+p_{E}\frac{\pi }{m}\right)u,$$
* which obviously is zero on*
L.* The power-conjugate output*
$y_{p}=\frac{\pi }{m}$
*is the velocity of the piston. One could also add an*
*extra*
*control Hamiltonian*
$\left(\frac{p_{S}}{T}+p_{E}\right)v$,* where*
$T=\frac{\partial U}{\partial S}$
*is the temperature, and*
*v*
*is the heat flow from an external heat source into the cylinder. The corresponding entropy conjugate output is*
$y_{e}=\frac{1}{T}$.

**Example 9** (Chemical reaction network continued [33])*.*
*Consider a chemical reaction network in*
*entropy representation*, *with the entropy*
*S*
*represented as a function*
S=S(E,x)
*of the vector of chemical concentrations*
*x*
*and energy*
*E*. *Then the Liouville submanifold describing the state properties of the reaction network is given as*
(67)$$L=\{\left(x,S,E,p_{x},p_{S},p_{E}\right)|S=S\left(E,x\right),p_{x}=-p_{S}\frac{\partial S}{\partial x}\left(E,x\right),p_{E}=-p_{S}\frac{\partial S}{\partial E}\left(E,x\right)\}$$
* with generating function*
$-p_{S}S\left(x,E\right)$
*and*
$\frac{\partial S}{\partial x}\left(E,x\right)=-\frac{\mu }{T},\frac{\partial S}{\partial E}\left(E,x\right)=\frac{1}{T}$. *The internal dynamics of the chemical reaction network are generated by the Hamiltonian*
(68)$$K^{a}=-p_{x}^{⊤}ZL\;\mathrm{Exp}\;\frac{-Z^{⊤}}{R}\frac{\partial S}{\partial x}\left(E,x\right)\;-p_{S}\frac{\partial S}{\partial x^{⊤}}\left(E,x\right)ZL\;\mathrm{Exp}\;\frac{-Z^{⊤}}{R}\frac{\partial S}{\partial x}\left(E,x\right)$$*Furthermore, the control Hamiltonian*(69)$$K^{c}=p_{S}\frac{\partial S}{\partial E}\left(E,x\right)+p_{E},$$*corresponds to a heat flow input, and an entropy flow conjugate output*$y_{e}=\frac{\partial S}{\partial E}{\left(x,E\right)}_{|L}$*equal to the reciprocal temperature. Another possible choice is*(70)$$K^{c}=p_{S}\frac{\partial S}{\partial x_{i}}\left(E,x\right)+p_{xi},$$*corresponding to material in/outflow of the i-th chemical species, with entropy flow conjugate output*$y_{e}=\frac{\partial S}{\partial x_{i}}{\left(E,x\right)}_{|L}$*equal to the chemical potential*$\mu_{i}$*of the i-th chemical species divided by the temperature T*.

**Example 10** (Heat exchanger continued [26])**.**
*The extensive variables are*
$S_{1},S_{2}$
*(entropies of the two compartments), and E (total internal energy). The state properties are described by the Liouville submanifold*
(71)$$L=\{\left(S_{1},S_{2},E,p_{S_{1}},p_{S_{2}},p_{E}\right)|E=E_{1}\left(S_{1}\right)+E_{2}\left(S_{2}\right),p_{S_{1}}=-p_{E}E_{1}^{\prime }\left(S_{1}\right),p_{S_{2}}=-p_{E}E_{2}^{\prime }\left(S_{2}\right)\},$$
*corresponding to the generating function*
$-p_{E}\left(E_{1}\left(S_{1}\right)+E_{2}\left(S_{2}\right)\right)$*, with*
$E_{1},E_{2}$
*as the internal energies of the two compartments. Denoting the temperatures*
$T_{1}=E_{1}^{\prime }\left(S_{1}\right),$$T_{2}=E_{2}^{\prime }\left(S_{2}\right)$*, the internal dynamics corresponding to Fourier’s law is given by the Hamiltonian*
(72)$$K^{a}=\lambda \left(\frac{1}{T_{1}}-\frac{1}{T_{2}}\right)\left(p_{S_{1}}T_{2}-p_{S_{2}}T_{1}\right),$$
*with*
λ
*Fourier’s conduction coefficient*.

## 5. Control by Interconnection

*Control by interconnection* is the paradigm of controlling a system by interconnecting it (through its inputs and outputs) to an additional *controller* system. The aim is to influence the dynamics of the original system by *shaping the dynamics* of the interconnected system by a proper choice of the controller system. Applied to port-thermodynamic systems, this means that given a plant thermodynamic system we interconnect it to a controller port-thermodynamic system such that in the closed-loop port-thermodynamic system the plant states converge to the desired set-point values. Port-thermodynamic systems can be interconnected, either by their power ports or by their entropy flow ports; cf. [26] for details. For example, the power port interconnection of two systems with variables
(73)$$\left(E_{i},S_{i},q_{i},p_{E_{i}},p_{S_{i}},p_{i}\right)\in T^{*}Q_{i},\;i=1,2,$$
is defined as follows. With the homogeneity assumption in *p* in mind, impose the following constraint
(74)$$p_{E_{1}}=p_{E_{2}}=:p_{E}$$

This leads to the summation of the Liouville one-forms $\alpha_{1}$ and $\alpha_{2}$ given by
(75)$$\alpha_{\mathrm{sum}}:=p_{E}d\left(E_{1}+E_{2}\right)+p_{S_{1}}dS_{1}+p_{S_{2}}dS_{2}+p_{1}dq_{1}+p_{2}dq_{2}$$
on the *composed space* $T^{*}Q_{1}\circ T^{*}Q_{2}$ defined as
(76)$$T^{*}Q_{1}\circ T^{*}Q_{2}:=\left\{\left(E,S_{1},S_{2},q_{1},q_{2},p_{E},p_{S_{1}},p_{S_{2}},p_{1},p_{2}\right)\right\}$$

Let the constitutive relations of the two port-thermodynamic systems be defined by the Liouville submanifolds $L_{i}\subset T^{*}Q_{i},i=1,2$. Then, the constitutive relations of the interconnected system are defined by the composition
(77)$$\begin{array}{ccc} L_{1}\circ L_{2} & := & \{\left(E,S_{1},S_{2},q_{1},q_{2},p_{E},p_{S_{1}},p_{S_{2}},p_{1},p_{2}\right)|\\ & & E=E_{1}+E_{2},\left(E_{i},S_{i},q_{i},p_{E_{i}},p_{S_{i}},p_{i}\right)\in L_{i},\;i=1,2\} \end{array}$$

Furthermore, consider the dynamics on $L_{i}$ defined by Hamiltonians $K_{i}=K_{i}^{a}+K_{i}^{c}u_{i},i=1,2$, where $K_{i}^{c}$ is the row vector of control Hamiltonians of system i,i=1,2. Additionally *assume* that the functions $K_{i}$ do *not* depend on the energy variables $E_{i},i=1,2$. Then $K_{1}+K_{2}$ is well-defined on $L_{1}\circ L_{2}$ for all $u_{1},u_{2}$. Next, consider the power conjugate outputs $y_{p1},y_{p2}$. By imposing *interconnection constraints* on the power port variables $u_{1},u_{2},y_{p1},y_{p2}$ satisfying the *power preservation* property
(78)$$y_{p1}^{⊤}u_{1}+y_{p2}^{⊤}u_{2}=0$$
one obtains an *interconnected port-thermodynamic system* with constitutive relations described by $L_{1}\circ L_{2}$. Similarly, interconnecting the inputs $u_{1},u_{2}$ to the entropy flow outputs $y_{e1},y_{e2}$ in such a way that
(79)$$y_{e1}^{⊤}u_{1}+y_{e2}^{⊤}u_{2}\geq 0,$$
leads again to a port-thermodynamic system.

A basic control problem concerns the *stabilization* of a system to a desired set-point value (*regulation*). How can we use control by interconnection to address this problem? Suppose we want to stabilize the system at some set-point value $\left(z^{*},S^{*}\right)$. If E(z,S) already has a strict minimum at $\left(z^{*},S^{*}\right)$ then one may asymptotically stabilize $\left(z^{*},S^{*}\right)$ by the interconnection with a *damper system* [32]. In fact, assume for simplicity of exposition that m=1 (scalar output $y_{p}$). Then, consider an additional linear damper system (cf. [26]), whose Liouville submanifold is given as
(80)$$L_{d}=\left\{\left(U_{d},S_{d}\right)|U_{d}=U_{d}\left(S_{d}\right),p_{S_{d}}=-p_{U_{d}}U_{d}^{\prime }\left(S_{d}\right)\right\},$$
with entropy as $S_{d}$, internal energy as $U_{d}\left(S_{d}\right)$, and $U_{d}^{\prime }\left(S_{d}\right)$ its temperature. The dynamics of this damper system are generated by the Hamiltonian (see [26])
(81)$$K=\left(p_{U_{d}}+p_{S_{d}}\frac{1}{U_{d}^{\prime }\left(S_{d}\right)}\right)du_{d}^{2}$$
(note the *quadratic* dependence on the input $u_{d}$), with power conjugate output $y_{d}=du_{d}$ (damping force). Then interconnect the plant port-thermodynamic system $\left(L,K=K^{a}+K^{c}u\right)$ to this damper system by setting
(82)$$u=-y_{d},\;u_{d}=y$$

This results (after setting $p_{U_{d}}=p_{E}$) in the interconnected port-thermodynamic system with total Hamiltonian given as
(83)$$K^{a}\left(E,S,z,p_{E},p_{S},p_{z}\right)-K^{c}\left(E,S,z,p_{E},p_{S},p_{z}\right)dy+\;\left(p_{U_{d}}+p_{S_{d}}\frac{1}{U^{\prime }\left(S_{d}\right)}\right)dy^{2}$$
with total energy $E\left(S,z\right)+U_{d}\left(S_{d}\right)$. This implies
(84)$$\frac{d}{dt}E\left(S,q\right)=-\frac{d}{dt}U_{d}\left(S_{d}\right)=-U_{d}^{\prime }\left(S_{d}\right)\dot{S_{d}}=-dy^{2}\leq 0$$

Hence, by an application of LaSalle’s invariance principle, the system converges to the largest invariant set within the set where the power conjugate output $y_{p}$ is zero. Note that y=0 corresponds to zero entropy production $\dot{S_{d}}=0$; in accordance with irreversible thermodynamics. If the largest invariant set where *y* is zero equals the singleton $\left(E^{*},S^{*},q^{*}\right)$ then asymptotic stability of $\left(E^{*},S^{*},q^{*}\right)$ results; for some limiting value $S_{d}^{*}$ of the entropy $S_{d}$ of the damper system; see also [32].

What can be done if E(S,z) does not have a strict minimum at $\left(S^{*},z^{*}\right)$ ? This can be approached via the (generalized) *Energy-Casimir* method; similar to the theory of control by interconnection for port-Hamiltonian systems, see e.g., [37,61]. Consider a port-thermodynamic system with Liouville submanifold $L\subset T^{*}Q$ with the generating function (in the energy representation) $-p_{E}E\left(S,z\right)$. A classical tool in the stability analysis of ordinary Hamiltonian dynamics is to consider additional *conserved quantities*; see e.g., [56,57,58]. In order to extend this idea to the present case, let us strengthen our assumption on $K^{a}$ by requiring that $\frac{\partial K^{a}}{\partial p_{E}}=0$*everywhere* on $T^{*}Q$; i.e., not just on L. Next, consider *additional* conserved quantities for the dynamics $X_{K^{a}}$ that are only depending on the extensive variables S,z; i.e., functions C(S,z) such that $\{K^{a},C\}=0$, where {·,·} is the standard Poisson bracket on $T^{*}Q$. Hence, also $\{K^{a},E+C\}=0$. Subsequently, note that [32]
(85)$$\alpha =p_{E}dE+p_{S}dS+p_{z}dz=p_{E}d\left(E+C\left(S,z\right)\right)+\left(p_{S}-p_{E}\frac{\partial C}{\partial S}\right)dS+\left(p_{z}-p_{E}\frac{\partial C}{\partial z}\right)dz$$

Hence, the transformation
(86)$$\left(E,S,z,p_{E},p_{S},p_{z}\right)\mapsto \left(E+C,S,z,p_{E},p_{S}-p_{E}\frac{\partial C}{\partial S},p_{z}-p_{E}\frac{\partial C}{\partial z}\right)=:\left(\tilde{E},\tilde{S},\tilde{z},{\tilde{p}}_{E},{\tilde{p}}_{S},{\tilde{p}}_{z}\right)$$
is a point transformation (that is, leaving the Liouville form invariant). Note that in the new coordinates the intensive variables $\frac{p_{S}}{-p_{E}},\frac{p_{z}}{-p_{E}}$ are transformed into the *new* intensive variables
(87)$$\begin{array}{c} \frac{{\tilde{p}}_{S}}{-p_{E}}=\frac{p_{S}-p_{E}\frac{\partial C}{\partial S}}{-p_{E}}=\frac{p_{S}}{-p_{E}}+\frac{\partial C}{\partial S}\\ \frac{{\tilde{p}}_{z}}{-p_{E}}=\frac{p_{z}-p_{E}\frac{\partial C}{\partial z}}{-p_{E}}=\frac{p}{-p_{E}}+\frac{\partial C}{\partial z} \end{array}$$
In these new coordinates the generating function for L in entropy representation is given by $\tilde{E}\left(S,z\right)=E\left(S,z\right)+C\left(S,z\right)$. Furthermore, since $\{K^{a},E+C\}=0$, the transformed Hamiltonian
(88)$$\tilde{K^{a}}\left(\tilde{E},\tilde{S},\tilde{q},{\tilde{p}}_{E},{\tilde{p}}_{S},\tilde{p}\right):=K^{a}\left(E,S,q,p_{E},p_{S},p\right)$$
satisfies $\{\tilde{K^{a}},\tilde{E}\}=0$. Hence, in the new coordinates we are back to the situation considered before: if $\tilde{E}\left(S,z\right)$ has a strict minimum at $\left(S^{*},z^{*}\right)$, then $\tilde{E}$ is a Lyapunov function for the dynamics restricted to L, and the equilibrium $\left(E^{*},S^{*},z^{*}\right)$ with $E^{*}=E\left(S^{*},z^{*}\right)$, is stable with respect to the dynamics on L.

Finally, note that the row vector $K^{c}$ in the new coordinates transforms to $\tilde{K^{c}}\left(\tilde{E},\tilde{S},\tilde{z},{\tilde{p}}_{E},{\tilde{p}}_{S},{\tilde{p}}_{z}\right)$, leading to the *transformed* power conjugate outputs
(89)$${\tilde{y}}_{p}:=\frac{\partial {\tilde{K}}^{c}}{\partial {\tilde{p}}_{E}},$$
and the *transformed* entropy flow conjugate outputs
(90)$${\tilde{y}}_{e}:=\frac{\partial {\tilde{K}}^{c}}{\partial {\tilde{p}}_{S}}$$

All this is illustrated by the stabilization of the gas-piston system in the following example.

**Example 11** (Regulation of gas-piston system)**.**
*Consider the gas-piston system (without a damper) with extensive variables*
(E,S,V,π), *as before. The constitutive properties of the system are given by the Liouville submanifold as in (*Equation 65*) with energy expression*
$E_{p}\left(S,V,\pi \right):=U\left(S,V\right)+\frac{\pi^{2}}{2m}$
*(‘p’ for plant). Without damping (*d=0)
*the dynamics are generated by the Hamiltonian*
(91)$$K=p_{V}\frac{\pi }{m}-p_{\pi }\frac{\partial U}{\partial V}+\left(p_{\pi }+p_{E}\frac{\pi }{m}\right)u_{p},$$
*with power conjugate output*
$y_{p}=\frac{\pi }{m}$
*(velocity of the piston). A scalar controller system with extensive variables*
$\left(E_{c},z_{c}\right)$
*is given by the port-thermodynamic system*
$\left(L_{c},K_{c}\right)$*, with energy*
$E_{c}=E_{c}\left(z_{c}\right)$*, and dynamics*
(92)$$K_{c}=\left(p_{c}+p_{E_{c}}E_{c}^{\prime }\left(z_{c}\right)\right)u_{c}$$
*with output*
$y_{c}=E_{c}^{\prime }\left(z_{c}\right)$*. The function*
$E_{c}\left(z_{c}\right)$
*is a design parameter, specifying the controller system*.
*The closed-loop system is obtained by the negative feedback (with v a new external input)*

(93)
$$u_{p}=-y_{c}+v=-E_{c}^{\prime }\left(z_{c}\right)+v,\;u_{c}=y_{p}=\frac{\pi }{m},$$

*together with*

(94)
$$E:=E_{p}+E_{c},\;p_{E_{p}}=p_{E_{c}}=:p_{E}$$


*This leads to the closed-loop Hamiltonian*

(95)
$$K=p_{V}\frac{\pi }{m}-p_{\pi }\frac{\partial U}{\partial V}+\left(p_{\pi }+p_{E_{p}}\frac{\pi }{m}\right)\left(-E_{c}^{\prime }\left(z_{c}\right)+v\right)+\left(p_{z_{c}}+p_{E_{c}}E_{c}^{\prime }\left(z_{c}\right)\right)\frac{\pi }{m}$$

*It is immediately seen that *$C\left(V,\pi ,z_{c}\right)=\Phi \left(V-z_{c}\right)$* for any function *Φ:R→R* is a conserved quantity. This motivates a consideration of new canonical coordinates *$\left(\tilde{E},\tilde{V},\tilde{\pi },{\tilde{q}}_{c},{\tilde{p}}_{E},{\tilde{p}}_{V},{\tilde{p}}_{\pi },{\tilde{p}}_{c}\right)$*, where*(96)$$\tilde{E}=E+\Phi \left(w=V-z_{c}\right),\;{\tilde{p}}_{V}=p_{V}-p_{E}\frac{\partial \Phi }{\partial w},\;{\tilde{p}}_{z_{c}}+p_{E}\frac{\partial \Phi }{\partial w},$$* while *$\tilde{V}=V,\tilde{\pi }=\pi ,{\tilde{z}}_{c}=z_{c},{\tilde{p}}_{E}=p_{E},{\tilde{p}}_{\pi }=p_{\pi }$*. In the new coordinates we compute*$\tilde{K}$*as*(97)$$\begin{array}{c} \tilde{K}=\left({\tilde{p}}_{V}+{\tilde{p}}_{E}\frac{\partial \Phi }{\partial w}\right)\frac{\tilde{\pi }}{m}-{\tilde{p}}_{\pi }\frac{\partial U}{\partial V}-\left({\tilde{p}}_{\pi }+{\tilde{p}}_{E}\frac{\tilde{\pi }}{m}\right){\bar{E}}_{c}^{\prime }\left({\tilde{z}}_{c}\right)+\\ \left({\tilde{p}}_{z_{c}}-{\tilde{p}}_{E}\frac{\partial \Phi }{\partial w}+{\tilde{p}}_{E}{\bar{E}}_{c}^{\prime }\left({\tilde{z}}_{c}\right)\right)\frac{\tilde{\pi }}{m}+\left({\tilde{p}}_{\pi }+{\tilde{p}}_{E}\frac{\tilde{\pi }}{m}\right)v, \end{array}$$*leading to the same power conjugate output *${\tilde{y}}_{p}=y_{p}=\frac{\pi }{m}$* (velocity of the piston). For any set-point *$V^{*}$* the functions *Φ* and *$E_{c}$* should be chosen in such a way that the function *$E\left(S,V,\pi ,\Phi (V-z_{c})\right)$* has a strict minimum at *$\left(S^{*},V^{*},\pi^{*}=0,z_{c}^{*}\right)$* for some value of *$S^{*}$* and state *$z_{c}^{*}$* of the controller system. As discussed before, this can be turned into asymptotic stabilization by additionally interconnecting the obtained closed-loop system with a damper system through the power port *$\left(v,{\tilde{y}}_{p}\right)$.

## 6. Conclusions and Outlook

Ever since the fundamental contributions of Gibbs and Maxwell to thermodynamics, *geometry* has played an essential role. Nevertheless, the development of the geometric theory of macroscopic thermodynamics still poses fundamental questions, especially when it comes to thermo*dynamics* instead of thermo*statics*. In this paper, the focus has been on quasi-Hamiltonian formulations based on the factorization of the irreversible entropy production, and on the contact-geometric approach using Liouville geometry. The first topic is intimately related to the GENERIC framework, as well as to the theory of irreversible port-Hamiltonian systems. With respect to the second topic, Liouville geometry offers a versatile framework for dealing with the general thermodynamic phase space, in particular by providing a unification of the energy and entropy representation. Although in this approach the *necessary* conditions for the dynamics on the thermodynamic phase space are clear, natural specifications of the dynamics are still somewhat lacking. In this regard, a *combination* of the quasi-Hamiltonian and GENERIC formulations with contact and Liouville geometry should be promising.

Finally, it should not be forgotten that thermodynamics started as an *engineering* subject (dealing with the efficiency of the steam engine). The interaction of thermodynamic systems with their surroundings is key to the theory. This has been demonstrated in this paper through a discussion of the definitions of the energy and entropy as storage functions with respect to supply rates corresponding to the thermal and the (mechanical) power port, and through the definition of port-thermodynamic systems. Furthermore, it naturally leads to *control* of thermodynamic systems, including the theory of ‘control by interconnection’.

## Figures and Tables

**Figure 1 entropy-25-00577-f001:**
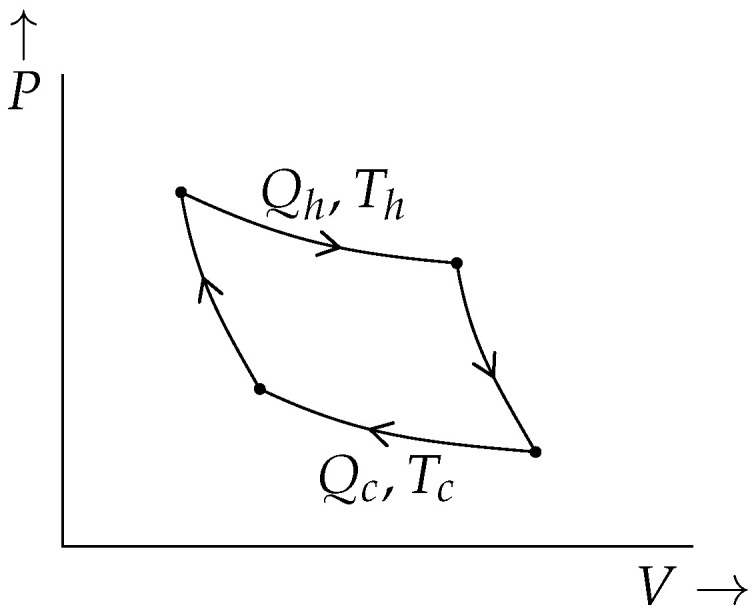
The Carnot cycle.

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
