# Peer review of "Geometric Modeling for Control of Thermodynamic Systems"

_entropy, 2023, doi:10.3390/e25040577_

Round 1

Reviewer 1 Report

The paper is an overview on geometric methods for modeling and control of irreversible thermodynamic systems. It addresses (port) Hamiltonian based formulations, specifically quasi-Hamiltonian, irreversible port-Hamiltonian and port-thermodynamic systems.

The paper is extremely well written and very pedagogic. The author succeeds in presenting in a clear and simple, but still precise and formal, manner the fundamentals of the mathematical formulation of irreversible thermodynamics. In my opinion this is a fundamental subject in science, and there are only very few papers that achieve to present and develop a discussion at the present level with such clarity. I enjoyed very much reading this paper and I definitively recommend it for publication and I celebrate the author for his work.

The list of references is quite complete but could be further completed in the following sections. 

Section 3.2 deals with stabilization and control of quasi-Hamiltonian formulations using the availability function. It would be good to mention some of the works of Alonso and Ydstie on stabilization of irreversible thermodynamic systems using the availability function, for instance [1]. In the same section some recent works on passivity based control using the availability function for quasi-Hamiltonian formulations generated by the total energy and the total entropy are given in [2] and [3], respectively. 

Section 4 and 5 deal with the definition and control of port-Thermodynamic systems. It would be interesting to (briefly) discuss the relation and differences with the definition and control of input-output contact systems, respectively [4] and [5]. I think one of the main benefits of defining thermodynamic models on the thermodynamic phase space is that it is possible to elegantly formulate geometric conditions that assures that a controlled (closed-loop) system is indeed again a thermodynamic system. 

Some typos I detected will reading the paper:

- line 110 missing reference.

- EQ: (16) missing spaces

- line 134 missing space

- EQ: (42) larger space between formulas 

- First sentence in Example 8, there is a "t" missing in the extensive

References:

[1] Antonio A. Alonso, B.Erik Ydstie, Stabilization of distributed systems using irreversible thermodynamics, Automatica, Volume 37, Issue 11, 2001, Pages 1739-1755.

[2] H. Ramirez, Y. Le Gorrec, B. Maschke and F. Couenne. On the passivity based control of irreversible processes: a port-Hamiltonian approach. Automatica, 2016. Volume 64, pages 105-111.

[3] H. Hoang, F. Couenne, C. Jallut, Y. Le Gorrec, The port Hamiltonian approach to modeling and control of Continuous Stirred Tank Reactors, Journal of Process Control, Volume 21, Issue 10, 2011,Pages 1449-1458,

[4] H. Ramirez, B. Maschke and D. Sbarbaro, Feedback equivalence of input–output contact systems, Systems & Control Letters, Volume 62, Issue 6, June 2013, Pages 475-481.

[5] H. Ramirez, B. Maschke and D. Sbarbaro. Partial stabilization of input-output contact systems on a Legendre submanifold. Automatic Control, IEEE Transactions on, vol. 62, no. 3, pp. 1431-1437, March 2017.

Author Response

NB: The changes in the paper are marked in RED.

The paper is an overview on geometric methods for modeling and control of irreversible thermodynamic systems. It addresses (port) Hamiltonian based formulations, specifically quasi-Hamiltonian, irreversible port-Hamiltonian and port-thermodynamic systems.

The paper is extremely well written and very pedagogic. The author succeeds in presenting in a clear and simple, but still precise and formal, manner the fundamentals of the mathematical formulation of irreversible thermodynamics. In my opinion this is a fundamental subject in science, and there are only very few papers that achieve to present and develop a discussion at the present level with such clarity. I enjoyed very much reading this paper and I definitively recommend it for publication and I celebrate the author for his work.

The list of references is quite complete but could be further completed in the following sections. 

Section 3.2 deals with stabilization and control of quasi-Hamiltonian formulations using the availability function. It would be good to mention some of the works of Alonso and Ydstie on stabilization of irreversible thermodynamic systems using the availability function, for instance [1]. In the same section some recent works on passivity based control using the availability function for quasi-Hamiltonian formulations generated by the total energy and the total entropy are given in [2] and [3], respectively. 

REPLY: Thanks for the evaluation, and thanks for suggesting to include these references. The references have been added, together with a few extra discussion lines (in RED) in Section 3.2 (between (39) and (40).

Section 4 and 5 deal with the definition and control of port-Thermodynamic systems. It would be interesting to (briefly) discuss the relation and differences with the definition and control of input-output contact systems, respectively [4] and [5].

REPLY: The connections with input-output contact systems have been elaborated in an extra paragraph (in RED) just before (65). Furthermore, I have included References [4] and [5]; thanks for pointing out.

I think one of the main benefits of defining thermodynamic models on the thermodynamic phase space is that it is possible to elegantly formulate geometric conditions that assures that a controlled (closed-loop) system is indeed again a thermodynamic system. 

Some typos I detected will reading the paper:

- line 110 missing reference.

- EQ: (16) missing spaces

- line 134 missing space

- EQ: (42) larger space between formulas 

- First sentence in Example 8, there is a "t" missing in the extensive

REPLY: Thanks for pointing out! All has been fixed.

References:

[1] Antonio A. Alonso, B.Erik Ydstie, Stabilization of distributed systems using irreversible thermodynamics, Automatica, Volume 37, Issue 11, 2001, Pages 1739-1755.

[2] H. Ramirez, Y. Le Gorrec, B. Maschke and F. Couenne. On the passivity based control of irreversible processes: a port-Hamiltonian approach. Automatica, 2016. Volume 64, pages 105-111.

[3] H. Hoang, F. Couenne, C. Jallut, Y. Le Gorrec, The port Hamiltonian approach to modeling and control of Continuous Stirred Tank Reactors, Journal of Process Control, Volume 21, Issue 10, 2011,Pages 1449-1458,

[4] H. Ramirez, B. Maschke and D. Sbarbaro, Feedback equivalence of input–output contact systems, Systems & Control Letters, Volume 62, Issue 6, June 2013, Pages 475-481.

[5] H. Ramirez, B. Maschke and D. Sbarbaro. Partial stabilization of input-output contact systems on a Legendre submanifold. Automatic Control, IEEE Transactions on, vol. 62, no. 3, pp. 1431-1437, March 2017.

REPLY: As mentioned above, all these (relevant!) references have been included in the revised version.

Reviewer 2 Report

The topic treated by the author belongs to a field which geometrizes classical thermodynamics with first steps already performed by Gibbs, Caratheodory and some others. In more recent times it was formulated within the field of contact geometry, see for example contributions by Hermann or Arnold. The author gives the topic an even more modern twist by using tools from control theory. The author is an expert in the field and the paper is well written. It is perhaps useful to add some remarks about the role of quasi static processes. I recommend publication in the journal Entropy.

Author Response

NB: Changes in the revised version have been marked in RED.

The topic treated by the author belongs to a field which geometrizes classical thermodynamics with first steps already performed by Gibbs, Caratheodory and some others. In more recent times it was formulated within the field of contact geometry, see for example contributions by Hermann or Arnold. The author gives the topic an even more modern twist by using tools from control theory. The author is an expert in the field and the paper is well written. It is perhaps useful to add some remarks about the role of quasi static processes. I recommend publication in the journal Entropy.

REPLY: Thanks for the positive evaluation. I have added an extra Remark concerning quasi static processes as Remark 1 (top of page 4; in RED).